# Intersection of Nutrition, Food Science, and Restaurant Research

**DOI:** 10.3390/nu17213490

**Published:** 2025-11-06

**Authors:** Christine Bergman, Yan Cao, Eunmin Hwang

**Affiliations:** Department of Food & Beverage and Event Management, University of Nevada Las Vegas, Las Vegas, NV 89154, USA; yan.cao@unlv.edu (Y.C.); eunmin.hwang@unlv.edu (E.H.)

**Keywords:** restaurants, nutrition, food science, bibliometric, review, menu labeling, obesity, food safety

## Abstract

Background/Objectives: Research on restaurants has traditionally emphasized business operations. Considering restaurants’ growing role in shaping dietary patterns and public health outcomes, this study aimed to map the scope, trends, and gaps in scholarly research addressing food-related aspects of restaurants, excluding business-oriented topics. Methods: A bibliometric analysis was conducted using the Web of Science and Scopus databases. Search terms encompassed multiple restaurant categories, including fast food, fast casual, casual dining, and fine dining. After screening, 956 peer-reviewed English-language journal articles were included. Descriptive performance metrics were calculated, and keyword co-occurrence analysis was conducted. Results: Findings revealed that nutrition-related studies dominate the literature, particularly research linking fast food consumption to obesity and the impact of menu labeling policies on consumer behavior. Food science research was comparatively limited and concentrated primarily on food safety and uses for degraded frying oil. The analysis also highlighted a strong research focus on fast food, while fast casual and fine dining restaurants were notably underrepresented. Conclusions: Future studies should move beyond short-term, cross-sectional designs and incorporate longitudinal approaches to better capture how policy interventions, such as menu labeling and reformulation incentives affect consumer food choices and restaurant offerings over time. Understanding how to reduce restaurants’ contribution to the incidence of diet-related noncommunicable disease risk factors such as obesity and hypertension will require research trials that jointly manipulate key factors such as economic (prices and incentives), structural (recipes, assortment, and operations), and behavioral (choice architecture). Research could also investigate strategies to reduce allergen risks by evaluating standardized training programs and integrated menu/POS disclosure systems. In addition, examination of consumer acceptance of sustainable ingredient substitutions and packaging methods is needed.

## 1. Introduction

Over the last few decades, research on restaurants has grown steadily, in tandem with the industry’s rapid development [1,2]. This expansion of the restaurant sector has significantly contributed to the economy, along with having impacts on humanity and the environment [3].

Bibliometric reviews of restaurant research have concentrated mainly on business-related topics. For example, Rodríguez-López et al. examined bibliometric research trends occurring in the fields of hospitality, leisure, sport, and tourism related to restaurants [4]. Another bibliometric analysis of peer-reviewed articles on restaurants examined study trends after first categorizing all business research into consumer behavior, organizational behavior, and finance [2]. A systematic review of restaurants’ business performance studied their research contexts, designs, and theoretical frameworks [5]. Bibliometric reviews have found dominant themes in restaurant business research to be around branding, digital transformation, corporate social responsibility, consumer behavior, consumer satisfaction, culture, and innovation, with minimal attention given to healthy eating [4,6]. While these reviews offer valuable insights into the business operations of the restaurant industry, little attention has been given to how food science and nutrition research intersect with restaurant practices.

Food science research contributes to product development, safety, shelf-life stability, and sensory quality of foods [7]. These are all critical to restaurant menu innovation, standardization, compliance with health regulations, and efforts to support food sustainability. Advances in food processing, ingredient substitution, and allergen control directly influence how restaurant operators meet consumer demands and regulatory requirements [8]. At the same time, nutrition research informs such activities as menu labeling, nutrient profiling, and dietary interventions. These are related to efforts to understand and address the rising rates of diet-related conditions and diseases [9]. Studies have consistently shown that foods consumed away from home are typically higher in kilocalories, saturated fat, sodium, and added sugars, and lower in essential nutrients, compared to home-prepared meals [10,11]. As dining has become more common, especially among younger people and those who live in urban areas, restaurants are increasingly positioned as both contributors to and potential mitigators of food sustainability and public health challenges [12].

This study presents a bibliometric analysis of research focused on the food-related dimensions of the restaurant industry, with no date restrictions applied in the selection of the articles. Specifically, it examines how scholarly work has addressed topics related to nutrition and food science within restaurant contexts. Business-related topics are intentionally excluded. The analysis tracks the evolution of scholarly interest over time, identifies key research themes, and highlights potential avenues for future investigation.

## 2. Literature Review

### 2.1. The Evolution of Food-Focused Research in the Restaurant Industry

The historical evolution of food-focused research in the U.S. reveals a long-standing connection between public health concerns and food environments. This relationship was brought to national attention with Upton Sinclair’s 1905 publication, The Jungle, which exposed unsanitary practices in U.S. slaughterhouses and triggered a wave of public outcry [13]. In response, the U.S. Food and Drug Administration and the U.S. Public Health Service launched the first voluntary “Restaurant Sanitation Program” in 1934. It introduced grade-based inspections and hygiene protocols that laid the foundation for today’s food safety standards [14].

As industrialization expanded throughout the 20th century, the food system was transformed. Advances in food science led to the mass production of consistent, affordable, and shelf-stable products [15,16]. These developments were instrumental in supporting the growth of the restaurant, and particularly the fast-food industry. Industrialization of the food supply also changed the nutritional profile of the foods commonly consumed by people in the developed and, increasingly, the less developed world [17]. The global rise in these ultra-processed products, which are often high in added sugars, refined carbohydrates, and fats, has prompted growing concern among nutrition researchers [18]. Increasing consumption of these foods in foodservice settings has been linked to higher rates of obesity, diabetes, and other chronic diseases [17,19].

Building on these historical developments, the following sections examine how food science and nutrition research have become increasingly relevant to the restaurant industry.

### 2.2. Nutrition and Restaurant Research

Nutrition research has traditionally focused on healthcare and community-based initiatives that are designed to guide dietary interventions for preventing and managing chronic disease conditions. In recent years, however, nutrition researchers have expanded their focus to include restaurant settings, whose foods have been associated with nutrition-related health risks as meals consumed away from home have become increasingly common.

In the U.S., food consumed away from home now provides a growing share of household kilocalories. The U.S. Department of Agriculture estimates that restaurants and similar establishments account for about 17% of all household caloric intake [20]. This trend is not limited to the U.S. In China, urban residents consumed roughly 18% of their dietary intake from meals prepared outside the home, while people in rural areas consumed 8% [21]. Similar increases have been documented in low- and middle-income countries like Brazil, India, and Peru, where the share of food consumed away from home has steadily risen [11,18]. A global review confirmed this trend, noting consistent growth in out-of-home food consumption across regions, particularly among young, urban, and higher-income groups [22]. Poor diet, in turn, contributes significantly to global mortality. The Global Burden of Disease Study estimated that suboptimal diets were responsible for approximately 11 million deaths worldwide in 2017, amounting to nearly one in five adult deaths, largely from cardiovascular disease, diabetes, and certain cancers [23]. At the same time, fast food consumption has surged globally, driven by rapid urbanization, economic growth, and aggressive marketing strategies. A recent systematic review found that increased availability and affordability of fast food have contributed to shifts in eating patterns and have been linked to rising rates of diet-related chronic diseases in both high-income and low- to middle-income countries [24]. Diets are changing with rising incomes and urbanization, specifically, people are consuming more animal-based foods, sugar, fats and oils, refined grains, and processed foods. This global shift, often referred to as the nutrition transition, represents a critical challenge for public health and food policy.

In response to these risks, governments have introduced policies and regulations targeting food environments, including the restaurant sector [25,26]. Regulatory strategies range from mandatory kilocalorie labeling to sodium reduction goals and public–private partnerships encouraging the availability of healthier food options. These efforts have elevated the importance of restaurants as both contributors to dietary risk and platforms for intervention. Nutrition researchers are now actively studying how restaurant practices can be influenced by and contribute to key areas of nutrition science. This literature review examines how five core subfields (i.e., public health nutrition, nutritional epidemiology, community nutrition, consumer food behavior, and foodservice systems) are shaping the connection between nutrition research and the restaurant industry.

Public health nutrition research has explored how policies such as menu labeling and food reformulation influence consumer behavior and menu quality. These policy tools aim to make it easier for diners to make healthier choices when eating out. A Cochrane review by Clarke et al. [27] found that kilocalorie labeling reduced energy ordered by about 11 calories per meal. While modest at the individual level, such reductions could have substantial impacts when applied across populations. Complementing this, Lachat et al. [11] reviewed restaurant-based health promotion strategies in Europe and found that interventions that combined multiple elements (e.g., staff training, labeling, and reformulation) were the most effective. Together, these findings suggest that nutrition policies are more likely to succeed when combined with structural and operational support for restaurants.

Nutritional epidemiology provides strong evidence connecting frequent restaurant dining with negative health outcomes. Studies have consistently shown that meals consumed away from home are typically higher in kilocalories, sodium, and saturated fats. Orfanos et al. [28], in a multi-country study across Europe, found that individuals who ate out more frequently consumed more energy and fewer essential nutrients. These findings have been echoed by other large-scale observational studies, which link high restaurant meal consumption with increased risk of obesity, heart disease, and metabolic disorders [11,29,30]. Such epidemiological data reinforces the need for targeted interventions in foodservice environments and justify the inclusion of restaurants in national nutrition surveillance and policy frameworks.

Community nutrition research has highlighted how local restaurants, especially in underserved areas, can be engaged to improve access to healthier food options. These efforts often focus on small, independent establishments that serve as primary sources of food in their neighborhoods. A systematic review by Valdivia Espino et al. [31] concluded that interventions combining point-of-purchase nutrition information with the introduction of healthier menu items were particularly successful. Building on this, Jostock et al. [32] found that owner-led changes, such as modifying cooking oils or reducing portion sizes, were both feasible and well-received in small restaurants. These findings suggest that community-based strategies, when designed collaboratively and tailored to local contexts, can lead to meaningful improvements in dietary behavior.

Consumer behavior research explores how diners perceive and respond to nutrition information provided in restaurant settings. Menu labeling, health icons, and promotional signage are commonly used strategies to influence customers’ food choices. According to a review by Krieger & Saelens [33], kilocalorie labeling tends to reduce the likelihood of people selecting high-kilocalorie items, though the effect is often modest. More recent analyses by Jostock et al. [32] suggest that these tools are more effective when combined with visually engaging and simple messaging. Together, these findings point to the need for well-designed communication strategies that not only inform, but also motivate healthier choices, especially in quick-service and high-volume dining environments.

Research on foodservice and nutrition has increasingly examined how policy interventions influence restaurant practices, particularly the production of healthier menu items. In response to growing public health concerns and regulatory initiatives, some large chain restaurants have introduced reformulated dishes aimed at improving the nutritional profile of their offerings. A scoping review by Rincón-Gallardo Patiño et al. [34] evaluated the impact of menu labeling policies on transnational restaurant chains. The review found evidence of reductions in energy content for newly introduced items among U.S.-based chains, though similar changes were not observed in other countries. Further insights are provided by Bleich et al. [35], whose systematic review of kilocalorie labeling and modified labeling interventions revealed mixed outcomes. While some restaurant operators made positive menu changes, the overall evidence was limited by study design flaws and underpowered analyses.

### 2.3. Food Science and Restaurant Research

Food science serves as a cornerstone of the restaurant industry, drawing on its core disciplines of microbiology, food engineering, chemistry, physics, and sensory analysis to inform and improve culinary practices, menu item development, and operational efficiency [8]. Traditionally associated with large-scale food production and processing, some food science scholarly research is explicitly performed for application in restaurant kitchens.

Food microbiology is the study of microorganisms and their interactions with food. This area of study directly underpins food safety practices in restaurants. This field develops techniques for identifying pathogens and designing packaging and processes that reduce microbial growth and survival. Its importance is underscored by the global burden of foodborne illness, with an estimated 600 million cases and 420,000 deaths annually, many linked to unsafe food handled in foodservice operations [36]. Innovations from food microbiology support applied systems like Hazard Analysis and Critical Control Points (HACCP), a globally recognized framework for hazard identification and control in foodservice [37,38]. In the U.S., the ServSafe^®^ program operationalizes HACCP principles through industry training on temperature control, cross-contamination, personal hygiene, and allergen management [39]. Restaurants operating at scale also adopt ISO 22000, a systems-level food safety standard that integrates HACCP with continuous improvement mechanisms [40].

According to the U.N. Environment Program, restaurants were responsible for 28% of global food waste [41]. A significant share of this waste comes from produce loss, with estimates showing that between 4% and 10% of food purchased by restaurants never reaches the customer [42]. The shelf life of fresh produce in foodservice operations is being extended through food engineering research that has resulted in the creation of modified atmosphere/modified humidity packaging. This technology works by controlling oxygen, carbon dioxide, and water vapor levels inside packages. It slows down respiration and moisture loss in fresh fruits and vegetables, helping to prevent spoilage and reduce food waste. Resealable films have been designed that protect the contents even after opening, allowing kitchens to store and reuse produce without any decrease in quality [43].

Molecular gastronomy demonstrates how food science drives creative innovation in restaurant cuisine by applying the principles of chemistry and physics to cooking. Techniques such as spherification, emulsification, and gelification enable chefs to modify the sensory properties of food. These methods provide precise control over flavor release, texture, and visual presentation [44,45]. These techniques not only enhance the multisensory dining experience, but also influence diners’ perception of value and creativity in high-end foodservice settings [46]. Sensory science studies have further demonstrated that controlled manipulation of texture and aroma through molecular gastronomy can significantly impact consumer satisfaction and emotional response [47]. Once limited to experimental kitchens, molecular gastronomy is now part of mainstream culinary education [48].

The research area called neurogastronomy focuses on how the brain creates and interprets flavor, emphasizing that such perception involves complex multisensory and cognitive processes rather than simple responses to food stimuli [49]. As discussed by Spence [50], research in this area explores how sensory cues, such as aroma, texture, visual presentation, and contextual information, such as pricing, may affect consumers’ neural responses during eating. For example, Chen, Papies, and Barsalou [51] identified a “core eating network” in the brain that integrates sensory and cognitive information to guide feeding behavior. Studies have also shown that brand and price cues can modulate neural activation in reward-related regions, leading participants to rate identical wines as tasting better when labeled with higher prices [52].

A term combining gastronomy and psychophysics is gastrophysics which is an area of research that investigates how environmental and contextual factors influence the perception and enjoyment of food [53]. It emphasizes that dining is a multisensory experience shaped not only by the chemical composition of food but also by cues found in the environment where the food will be consumed [50]. Research in this area has shown, for example, that the color and shape of tableware can alter perceived sweetness or saltiness [54], and that background music can modulate taste perception and emotional responses to food [55]. Similarly, artistic plating has been found to increase both enjoyment and willingness to pay, even when the ingredients remain the same [56].

Sensory science methodology is used to optimize the sensory attributes of food while controlling ingredient costs, extending shelf life, and enhancing nutritional value, drawing on data from both consumer sensory evaluations and trained sensory panels. This scientific approach enables foodservice operators to evaluate how foods are perceived in terms of taste, texture, aroma, and appearance, allowing for informed decisions about product formulation, ingredient sourcing, and preparation methods. Civille and Carr [57] emphasize that sensory evaluation techniques help ensure food products meet customer expectations and maintain consistent quality. Kumpulainen et al. [58] demonstrated this by assessing consumer responses to different levels of lettuce freshness in a restaurant setting, finding that diners preferred freshly cut lettuce over packaged options due to superior texture and color. This shows that small differences in product quality can significantly influence restaurant customer satisfaction. In a different context, Hicks-Roof et al. [59] used sensory evaluation to assess college students’ acceptance of healthier menu items made with whole-grain sorghum. Although sorghum scored slightly lower in texture than white rice, overall acceptance was comparable, highlighting how sensory testing can inform the development of nutritious alternatives that are likely to be accepted by diners.

## 3. Methods

### 3.1. Databases and Search Terms

To achieve comprehensive coverage of relevant literature, data was extracted from the Web of Science (WoS) and Scopus databases [4]. WoS is recognized for hosting high-quality research, and Scopus is valued for its extensive inclusion of social science research articles [60]. Titles and keywords were searched in both databases with the following Boolean query: (multi-unit OR chain OR franchise OR “fast food” OR “quick service” OR “quick serve” OR “fast casual” OR “quick casual” OR “casual dining” OR “fine dining” OR fine-dining OR “upscale dining”) AND (restaurant OR foodservice). The National Restaurant Association categorizes restaurants as fast food, fast casual, casual dining, or fine dining [61]. Synonyms for these terms, such as “quick service” were included to ensure broader search results.

### 3.2. Inclusion and Exclusion Criteria and Data Screening Process

The data extraction process involved two steps and is reported in a PRISMA flowchart (Figure 1). First, only peer-reviewed, English-language journal articles were considered, yielding 1127 articles from WoS and 2313 from Scopus, amounting to 3440 articles. Repetitive articles were manually removed using reference manager, Zotero (Version 7), resulting in 1995 articles for further review. Next, titles, keywords, and abstracts were assessed to determine each article’s suitability for the study. Only articles addressing nutrition or food science-related aspects of the restaurant industry were included. Articles related to the business themes of the restaurant (i.e., organizational behavior, consumer behavior, and finance) sector or lacking full text were not selected [2]. However, articles with business topics focused on nutrition or food science, such as pricing and marketing effects on food choice, were included. Ultimately, 956 journal articles were used for this bibliometric analysis.

### 3.3. Data Analysis

A bibliometric analysis was conducted to explore the progression and structure of nutrition and food science-related restaurant research [62]. Both descriptive performance analysis and keyword co-occurrence analysis were conducted. The descriptive performance analysis was performed with total citation scores and citation per year index using Harzing’s Publish or Perish (Version 8) [63]

VOSviewer (version 1.6.20) was employed to conduct a scientific mapping of the bibliometric data [62]. Specifically, it was used to identify key topics and research gaps within the collected literature. Through keyword co-occurrence analysis, the software visualized relationships among keywords based on their frequency and co-location within the publications [64]. Network visualization maps were generated for documents categorized as nutrition (*n* = 824) and food science-focused (*n* = 132). Only keywords that appeared at least five times (*n* = 5) were included in these maps.

To uncover themes within the nutrition and food science datasets, we reviewed the titles, keywords, and abstracts of each article to extract relevant explanatory variables. These variables were refined by merging similar or overlapping terms into broader thematic categories. After categorizing the explanatory variables, we quantified the number of articles linked to each category and determined their percentage relative to the number of articles in the related database (i.e., nutrition and food science).

## 4. Results

### 4.1. Article Numbers by Year and Restaurant Category

All articles identified through the key search terms were analyzed to determine the yearly publication trends in the Scopus and Web of Science Databases. The annual publication rate remained relatively low for the nutrition and food science-related papers until the early part of the 21st century (Figure 2 and Figure 3). As depicted in Figure 2, the domain of nutrition related to the restaurant industry has been extensively studied, with 824 articles published, compared to food science (Figure 3), which includes 132 articles.

### 4.2. Journal Outlets

Among the journals that have published the most articles related to nutrition in restaurants, Public Health Nutrition ranks first, contributing 80 of the 824 analyzed articles (9.71%). It is followed by the American Journal of Preventive Medicine with 49 articles (5.95%), Nutrients with 35 articles (4.25%), and Appetite with 31 articles (3.76%) (Figure 4). Other notable journals include the Journal of the Academy of Nutrition and Dietetics with 28 articles (3.40%), BMC Public Health with 26 articles (3.16%), and the International Journal of Behavioral Nutrition and Physical Activity with 25 articles (3.03%). Additionally, Health and Place contributed 23 articles (2.79%), the International Journal of Environmental Research and Public Health had 20 articles (2.43%), and the American Journal of Public Health accounted for 19 articles (2.31%).

Several of these journals are focused on Public Health, while the others have Public Health as a category of their focus. Four of the journals follow a gold open-access publishing model, including Nutrients, BMC Public Health, International Journal of Behavioral Nutrition and Physical Activity, and International Journal of Environmental Research and Public Health. The remaining journals follow a subscription-based model but offer authors the option to make individual articles open access for a fee, classifying them as hybrid journals.

Among the top 10 journals that published the highest number of food science articles, the International Journal of Food Microbiology and the Journal of Food Protection each contributed 6 out of the 132 analyzed articles (4.55%), followed by Eurosurveillance with four articles (3.03%) (Figure 5). Other notable journals include the Japanese Journal of Infectious Diseases with four articles (3.03%), Epidemiology and Infection with three articles (2.27%), Food Control with 3 articles (2.27%), and Public Health with three articles (2.27%). Additionally, the American Journal of Potato Research and Bioresource Technology each published two articles (1.52%). There are also 13 other journals, each contributing two articles (1.52%).

A large portion of these journals are focused on topics that include food safety such as International Journal, Food Microbiology, Journal of Food Protection, and Eurosurveillance. Among these, the Journal of Food Protection, Eurosurveillance, Japanese Journal of Infectious Diseases, and Epidemiology and Infection follow a gold open-access publishing model. The remaining journals are categorized as hybrid journals.

### 4.3. Most Cited Articles

#### 4.3.1. Highly Cited Articles in Nutrition Research Related to Restaurants

In the nutrition and restaurant research articles, all ten of the articles with the greatest number of citations focus on fast-food restaurants (Table 1). These top-cited studies collectively examine factors contributing to obesity, particularly the associations of fast food consumption with various demographics such as adults, adolescents, and women, as well as socio-economic factors like low income, behavioral, and psychosocial variables [65,66]. These articles highlight the correlation between fast-food restaurant use and increased energy intake, poorer nutrient profiles, and higher rates of overweight and obesity. Several studies examined the proximity of fast-food restaurants to where adolescents live and attend school, as well as the spatial clustering of these outlets around schools as a potential contributor to an obesogenic environment.

#### 4.3.2. Highly Cited Articles in Food Science Research Related to Restaurants

The most cited articles that examined food science and restaurant topics focus on food safety, food additives, and food environmental impact (Table 2). The studies that investigated aspects of food safety included a molecular epidemiology study of norovirus outbreaks in the U.S. [67], the presence of fecal coliforms and E. coli in ready-to-eat salads irrigated with untreated sewage water [68], and the identification of tuna in sushi using DNA barcoding techniques [69]. A review examined the areas of the restaurant industry that need improvement in the implementation of the HACCP system [70], while another study examined the levels of potentially harmful anthropogenic compounds in tap water and beverages from fast-food franchises [71]. The continuing controversy over the safety of using monosodium glutamate in restaurant food and the ability of green tea extract to decrease the production of heterocyclic amines during frying was examined [72]. Finally, greenhouse gas emissions from beef and chicken production were assessed as part of efforts to understand the environmental impact of restaurant food consumption [73,74].

### 4.4. Restaurant Research Focused on Nutrition Topics

Below is a map of the keyword co-occurrence in nutrition research that is focused on restaurants (Figure 6). Out of the 3868 keywords associated with these research studies, a co-occurrence analysis revealed 67 keywords that met the threshold of a minimum number of occurrences (*n* = 50). These keywords were grouped into three thematic clusters, centered on the theme of fast food, including restaurant and public health (red), obesity and neighborhood (green), and demographics and methodology type (blue).

The red cluster, positioned on the left side of the map, centers on restaurant topics related to food nutrient composition, consumer decision-making, and public health policy. Prominent nodes such as calorie, nutritional value, and sodium are among the largest in this cluster. They reflect the high volume of research addressing the calorie and sodium content of restaurant foods and the nutritional labeling of menus in restaurant settings. The frequent appearance of nutrition policy, healthcare policy, and public health indicates there has been a large amount of research attention placed on the effects of regulatory efforts focused on influencing restaurant offerings and consumer choices in them. Additionally, the presence of the term food preference suggests an interest in the psychological and behavioral aspects of food choice, particularly how consumers balance taste preferences with nutrition-related information when ordering from restaurant menus.

The blue cluster, located in the upper central region of the keyword co-occurrence map, reflects a concentration of terms related to study design characteristics and the demographic profiles of participants in research on nutrition and restaurant contexts. Frequently occurring keywords such as adult, female, male, young adult, middle-aged, preschool child, and adolescent indicate that the literature heavily focuses on human populations across various life stages. The smaller nodes, aged and child, indicate that older adults and very young children are less frequently represented in this body of research.

Methodologically, this cluster includes keywords like human, health survey, cross-sectional study, and human experiment. These terms collectively point to a heavy reliance on short-term observational designs, surveys, and human experimental trials. The central placement of human and its extensive linkages across clusters reflect the dominant role of human-subject research in this field. Notably absent from the map is any distinct node representing longitudinal designs, underscoring a significant gap in the methodological landscape. This absence suggests that the field lacks sustained attention to the long-term health effects of foodservice consumption patterns and related interventions.

The green cluster, located on the right side of the co-occurrence map, represents a concentration of research focused on health outcomes and the broader environmental and social contexts in which restaurant-related nutrition issues occur. Large nodes such as obesity, body mass index, prevalence, and risk factors indicate that much of the literature connects restaurant food consumption to concerns about trends in excess weight at the population level. The central position of the obesity node within this cluster highlights its role as a primary focal point and a key linking concept to other areas of study in the network. This indicates that less research has focused on the association between restaurant food consumption and other chronic health problems, such as hypertension.

Also prominent in this cluster are keywords that reflect contextual influences, including socioeconomics, residence characteristics, neighborhood, and food supply. Their strong co-occurrence with health-related terms suggests that the literature frequently examines how factors such as income, community characteristics, and local food environments intersect with nutrition and health outcomes in relation to foodservice venues. The proximity of these environmental terms to health outcome keywords underscores an interest in learning how place-based and social conditions may shape dietary patterns.

Through inductive analysis of nutrition-focused articles within the restaurant research dataset, a set of thematic categories emerged based on explanatory variables extracted from each study (Figure 7). These themes were developed through a thorough review of article titles, keywords, and abstracts, with overlapping concepts consolidated into broader areas of focus.

The most frequently represented theme was *fast food exposure (foodscape)*, accounting for 29% of the articles. Research in this area emphasized the influence of the external food environment, particularly the density, accessibility, and spatial distribution of fast-food outlets on dietary patterns and public health outcomes. *Nutrition labeling* was another prominent theme, representing 11% of the articles. These studies examined the implementation and impact of kilocalorie and other nutritional information provided on menus. A common focus was on whether such labeling impacts consumer choices and supports healthier eating in restaurant settings. *Food allergens* also accounted for 11% of the dataset, highlighting the challenges consumers face when managing food allergies in restaurant environments. This body of research explored the difficulties individuals encounter in identifying safe menu items, communicating their dietary restrictions, and feeling confident that their needs will be accommodated when dining out.

Other themes included *food kilocalorie and nutrient content* (5%), which centered on the analysis of the nutritional quality of menu items in restaurant settings. These studies often quantified kilocalorie content, macronutrient distribution, and the presence of key nutrients or unhealthy components such as saturated fat, sodium, and added sugars. This theme aimed to provide insights into how restaurant food contributes to overall dietary intake and public health nutrition concerns.

*Fast food consumption* (4%) encompassed research exploring individual behaviors and demographic patterns associated with frequent consumption of fast food. This included studies investigating predictors such as age, income, and lifestyle factors, as well as the relationship between fast food intake and health outcomes like obesity or diet-related chronic diseases.

Several smaller but relevant themes were also identified. *Government nutrition policy* (3%) focused largely on the effects of kilocalorie menu labeling policies. Studies in this area examined whether providing kilocalorie information on menus influenced customer purchasing behavior, led to healthier food choices, or encouraged restaurants to reformulate or modify menu offerings to reduce kilocalorie content. A smaller number of studies also explored the impact of sugar-sweetened beverage taxes, assessing whether these fiscal measures affected consumer purchasing patterns and contributed to broader public health goals. *Food marketing effects* (3%) included research on the association between fast food marketing, particularly on television, and increased fast food consumption, especially among children and adolescents. In addition, studies examined in-restaurant advertising practices, such as signage and promotional strategies encouraging combo meals and upsizing, and how these tactics influenced customer purchasing behavior and overall kilocalorie intake. Finally, *psychosocial and environmental determinants of food choice* (2%) addressed individual and contextual factors, such as social norms, stress, convenience, and time constraints that affect how people make food decisions in restaurant environments. Of the articles, 32% were categorized as *other*, capturing studies whose explanatory variables did not align with the primary themes identified.

### 4.5. Restaurant Research Focused on Food Science Topics

The keyword co-occurrence analysis revealed three major clusters representing distinct thematic areas of research at the intersection of restaurants and food science (Figure 8). The network visualization shows keyword groupings that highlight the scope and relationships between key topics in the literature. Out of the 1994 keywords, a co-occurrence analysis revealed 61 keywords that met the threshold for the minimum number of keyword occurrences (*n* = 5). These keywords were grouped into three thematic clusters: red, blue, and green.

The red cluster located on the right side of the co-occurrence map appears to be focused on topics related to public health surveillance in the context of restaurants and foodservice establishments. A striking feature of this cluster is the presence of the keywords feces and feces analysis. In restaurant-related food science research, these terms signify the microbiological testing of stool samples from individuals affected by suspected foodborne outbreaks linked to food outlets. Keywords such as human, adult, male, female, and age group categories reflect studies that examine how foodborne illness affects different segments of the population, often identifying vulnerable groups such as young children and older adults. Finally, the inclusion of the keyword U.S. indicates that a significant portion of the research studies were focused on that country.

The green cluster, which is located at the bottom of the co-occurrence map, focuses on operational and food-handling aspects of the restaurant industry, in particular within the fast-food industry. Core keywords include hygiene, food safety, restaurant, catering service, food processing, cooking, and fast food. These terms indicate studies exploring hygiene practices, safe food preparation methods, and foodservice management. The inclusion of the commodity-specific keywords meat, cattle, and Solanum tuberosum (potato) points to research examining the handling of raw ingredients and their safety implications.

The blue cluster located on the top of the co-occurrence map centers on laboratory-based food science investigations, with keywords such as nonhuman, microbiology, Escherichia coli, Staphylococcus aureus, polymerase chain reaction, and bacterium identification. This cluster reflects research focused on pathogen detection and classification in foodservice contexts. The presence of the keywords seafood, fish, and animal suggests an emphasis on specific food sources that are common transmitters of foodborne illness. This body of work primarily addresses the technical and analytical aspects of food safety in restaurants and catering services.

Food science–related articles focusing on restaurant research were analyzed using an inductive approach to uncover their explanatory variables. These variables were extracted from article titles, abstracts, and keywords, and grouped into thematic categories. The resulting themes capture major scientific priorities in foodservice environments (Figure 9).

The most prominent theme was *frying oil quality and safety*, comprising 14% of the articles. Studies in this category examined the chemical degradation of oils during repeated use at high temperatures and the formation of harmful compounds, including acrylamide, acrolein, polymerized triglycerides, and 4-hydroxy-2-nonenal. These articles explored the effects of oil interceptors, various frying practices in restaurants, and uses for spent oils [75,76]. The *sustainability of food and use of food waste* theme accounted for 8% of the studies and addressed strategies such as converting food waste into animal feed or biodiesel and applying life-cycle assessment tools to restaurant operations [73,77,78]. *Food safety (microbial)* represented 7% of the dataset and primarily focused on the contamination risks associated with ingredients in restaurants and carried by employees. Studies investigated the presence of foodborne pathogens such as *Escherichia coli*, *Salmonella* spp., and Clostridium perfringens, which are commonly introduced through poor hygiene, cross-contamination, or improper temperature control [79,80]. Research also evaluated preventive measures in foodservice environments, such as kitchen sanitation practices and cold chain management in minimizing microbial hazards [81,82,83].

*Food quality and fraud* accounted for 5% of the studies and addressed issues of product mislabeling and ingredient authenticity [69,71]. For example, one study used DNA barcoding to detect mislabeled tuna species in sushi [84]. The *equipment suitability* theme (3%) explored the reliability and effectiveness of kitchen appliances used in commercial foodservice. These articles investigated how well kitchen equipment was able to maintain temperature control and supported safe storage and transportation [70,81,85]. Another 3% of the studies focused on *food additives*, examining substances such as preservatives and flavor enhancers that are used in restaurant settings and their health and sensory property effects [86,87]. For example, Taliaferro [88] reviewed the literature related to the effects of monosodium glutamate consumption from foods in Chinese restaurants.

A large share of the dataset (*n* = 60%) of articles was categorized in the *other* theme of explanatory variables as they did not fit into of the previously discussed themes. For example, these studies included one on wastewater treatment in foodservice operations [89], trace element contamination in beverages sold by fast-food chains [71], and applications of artificial intelligence in developing personalized restaurant menus [90].

### 4.6. The Breadth of Restaurant Research in the Nutrition and Food Science Domains

We conducted a manual review of the articles to learn how the five categories of restaurants intersect with the two major food-related domains: nutrition and food science (Table 3). The review identified that several intersections between restaurant categories and food domains have not been examined. Specifically, fine dining restaurants have not been studied in the context of the nutrition field. Similarly, there is a gap in research addressing nutrition and food science issues within the fast casual restaurant sector.

## 5. Discussion

The current study is the first bibliometric review that holistically investigates nutrition and food science research focused on the restaurant industry. Substantially more nutrition than food science-related research on restaurants was identified in this review. One likely reason for this imbalance is that much of the food science work conducted within restaurant companies is proprietary, kept in-house to maintain competitive advantage, and therefore not published in scholarly journals. Also, food scientists tend to publish less restaurant-focused research than nutrition scientists because those working in the academe have access to little financial support for these projects [91]. By contrast, nutrition-related restaurant research, particularly studies on the effects of menu labeling, consumer food behavior, and health outcomes, aligns directly with public health agendas, making it more attractive for government and foundation funding and more likely to be widely disseminated.

### 5.1. Restaurant Research Focused on Nutrition Topics

A significant portion of the nutrition and restaurant studies in this bibliometric analysis were focused on public health topics, with findings from both classic and contemporary research consistently showing that frequent dining out is associated with higher energy intake and less favorable nutrient profiles. Much of this work centered on increased exposure to fast food being related to a higher incidence of noncommunicable chronic diseases (e.g., type II diabetes and hypertension). Research analyzing the nutrient composition of restaurant offerings further documents high energy density and elevated levels of nutrients of concern, such as sodium, sucrose, and saturated fat, in many menu items [11,28]. This body of work emphasizes how outlet density, accessibility, and spatial distribution influence dietary patterns and overall population health. Regions where high concentrations of quick-service and convenience outlets overwhelm access to healthier options are sometimes called food swamps [91]. Consistent with this framing, highly cited studies linked frequent fast food consumption and proximity to these outlets (including those near schools) with higher energy intake, poorer nutrient profiles, and elevated risks of overweight and obesity (e.g., [65,92,93,94,95]). In parallel, research on food swamps finds that these environments predict obesity rates at least as well as, and in some analyses better than, limited-access food deserts, reinforcing the importance of restaurant-dense foodscapes in population health [96,97]. Building on this, the most-cited study in the set [91] highlights that greater restaurant availability and lower prices for meals away from home are associated with higher obesity over time. The study further connects these trends to societal shifts, such as rising time pressure and increased labor-force participation, that have boosted demand for convenient, ready-to-eat foods and, in turn, fast food consumption.

A significant portion of the literature addressing nutrition and restaurants focused on the mix of policies and regulations that have been adopted to address the obesity epidemic. In response to these, studies most often assessed point-of-purchase levers, especially menu kilocalorie labeling in chain restaurants; reviews and evaluations consistently report greater kilocalorie awareness and small, meaningful reductions in the energy content of food ordered, with effects varying by setting and population studied [66,98,99,100,101]. Evidence also points to supply-side responses that followed labeling policies. Specifically, some large chains introduced lower-energy items, indicating that some product reformulation had occurred [34,102]. Fiscal policies that increased taxes on sugar-sweetened beverages aimed to reduce purchases of high-calorie drinks and reduce overall energy intake. These types of policies were found by studies to result in significant pass-through to prices and sales declines in several global regions [103,104,105]. Some jurisdictions have implemented zoning and planning restrictions on new fast-food outlets, especially those near schools and in high-obesity areas [94,106,107,108,109,110].

Building on the previous discussion of policies and regulations related to nutrition in restaurants, it is important to note that while allergens in restaurant settings were included in this bibliometric review, they accounted for a far smaller share of research output compared to studies on the nutrient quality of restaurant foods. None of the top-cited nutrition and restaurant articles focused on allergens, underscoring the lower level of scholarly attention in this area. The allergen-related research that does exist primarily examines three core areas: (1) variability in policies and regulations across jurisdictions and their implications for consumer protection; (2) gaps in restaurant staff knowledge, attitudes, and practices regarding allergen management; and (3) communication challenges between staff and customers when conveying allergen information. Key research questions have included how effectively regulations are implemented, whether staff training translates into safer practices, and what barriers diners face in obtaining reliable allergen information. Studies have found inconsistent implementation of allergen management requirements, variability in staff preparedness, and persistent communication barriers [111,112,113]. Diners often report difficulty interpreting menu allergen information and uncertainty about whether their requests will be met safely [114]. Systematic reviews emphasize the need for standardized allergen disclosure formats, consistent training, and stronger enforcement mechanisms [115,116]. Collectively, this evidence highlights that although allergen-related policies exist, their practical impact remains limited.

Across the nutrition and restaurant literature, the most common methodologies were cross-sectional surveys, short-term experimental studies, and secondary data analyses of menu offerings or sales records. These designs dominate much of the research on menu labeling, nutrient composition, and consumer behavior, offering valuable descriptive and correlational insights but limited ability to assess long-term effects. In contrast, longitudinal cohort studies, randomized controlled trials in real-world restaurant settings, and natural experiments tied to policy implementation were used far less frequently. Longitudinal studies are essential in this area because they can determine whether the effects of menu labeling and similar interventions are sustained over time, helping to distinguish between temporary behavioral shifts driven by novelty or external cues and lasting changes in dietary decision-making patterns.

### 5.2. Restaurant Research Focused on Food-Science Topics

The restaurant industry employs food scientists to ensure that products meet customer expectations, maintain consistent sensory and nutritional properties across locations, and are produced safely and cost-effectively. Because the greatest risks to both consumers and operators arise when food safety systems fail, it is not surprising that many of the restaurant-focused food science studies in this bibliometric analysis concentrated on this topic. Failures in food protection can occur when microbial pathogens survive due to improper cooking or cross-contamination, when chemical hazards accumulate from unsafe processing or packaging practices, or when physical contaminants such as mislabeled ingredients or foreign objects reach consumers.

Within this literature, explanatory variables clustered around the three major categories of food safety: microbial, chemical, and physical hazards. Microbial food safety studies examined pathogens such as *Escherichia coli*, *Salmonella* spp., Staphylococcus aureus, and norovirus, reflecting the public health urgency of preventing outbreaks linked to foodservice operations [67,68]. Chemical food safety studies focused on degradation of frying oils and the formation of toxic by-products such as acrylamide, acrolein, and 4-hydroxy-2-nonenal, as well as food additive concerns such those related to monosodium glutamate [75,88,117]. Physical food safety research investigated authenticity and fraud, such as DNA barcoding to detect seafood mislabeling [69].

Although food safety dominated the food science research identified in this bibliometric review, a smaller set of studies focused on the integration of food science and sustainability, addressing issues directly relevant to restaurant operations. These topics are important because restaurants face increasing pressure to reduce environmental impacts, control costs, and meet consumer demand for sustainable practices. Research on the reuse of frying oils discarded by restaurants explored their conversion into biodiesel. Thus, presenting restaurant operators with opportunities to lower disposal costs and contribute to renewable energy efforts [78,118]. Complementing this line of inquiry, other work examined the potential to reuse frying oil in the development of biodegradable lubricating greases [119].

Studies on wastewater management assessed grease interceptors and microbial treatments to reduce fats, oils, and grease before disposal, helping restaurant operators avoid sewer blockages, regulatory fines, and negative environmental impacts [76,89]. Similarly, life-cycle assessments of protein source production provided insights into greenhouse gas emissions, land use, and water use, which are needed to inform sustainability reporting [73].

### 5.3. Research Gaps and Suggestions for Future Nutrition and Restaurant Research

Understanding how to reduce restaurants’ contribution to the incidence of diet-related noncommunicable disease risk factors (e.g., obesity and hypertension) will require that research trials be run that jointly manipulate prices and incentives (economic), recipes/assortment/operations (structural), and choice architecture (behavioral). This is because single-lever policies such as kilocalorie labeling of menus have produced only small to modest average changes in consumer purchasing behavior [120,121]. Such future research could focus on restaurants with multiple units in the fast food and fast casual sectors, as more people eat food from these establishments than other types of restaurants. This research should also account for the demographic characteristics of the customers being studied, as cultural backgrounds and regional contexts may lead to differing responses to these interventions.

On the economic side, studies could randomize menu-level price architectures that make healthier meal builds the better deal at the exact moment of choice (e.g., value meals that price fruit/vegetable sides and water at parity or at cost; small price surcharges for energy-dense swaps). Prior work shows that relative prices reliably shift demand for promoted healthier options, but most evidence comes from retail or single-lever tests [122,123]. Restaurant-specific trials could measure not only nutrition outcomes (kilocalories and sodium per ticket) but also effects on contribution margin, average check, and traffic by daypart to confirm commercial viability.

For the structural component, trials could implement stepwise sodium and energy reformulation in high-contribution items (e.g., stocks/bases, sauces, breads) aligned with the U.S. Food and Drug Administration’s voluntary sodium targets, using gradual reductions and salt-replacement strategies that are validated by sensory evaluation studies [124,125]. Because supply costs can be a barrier to this type of restaurant change, studies could offer limited-time subsidies to the restaurants for produce and plant proteins to hold plate costs steady. This coupling research will test whether upstream affordability (economic) amplifies the impact of recipe changes (structural).

On the behavioral front, field experiments could embed healthy defaults and other menu architecture directly into ordering flows. For example, restaurants can make healthier choices easier by setting lower-calorie sides as the default in combo meals, placing them in top menu positions, and giving guests simple prompts to choose smaller portions. Evidence from restaurants and online food environments shows that menu defaults, positioning, and portion prompts can reduce energy ordered, but effectiveness varies by context and strength of preference, underscoring the value of testing them alongside pricing and reformulation [126,127,128]. Because digital ordering has become ubiquitous, trials could also evaluate app-based nudges (e.g., reduced-salt defaults, “lighter build” toggles, personalized just-in-time prompts) that have previously shifted orders toward lower-sodium and healthier choices on meal-delivery platforms [129,130,131].

Critically, the unit of randomization and measurement must reflect restaurant realities. Multi-arm, multi-component trials can randomize outlets to: (1) pricing only; (2) pricing + reformulation; (3) pricing + reformulation + defaults/digital nudges; and (4) usual practice. Outcomes should be captured from point of sale (POS) and delivery systems (item-level sales mix, average check, contribution margin, ticket time, throughput, refund/complaint rates), linked to nutrition metrics (kilocalories and sodium per item/ticket), and customer experience (satisfaction, re-order rates). This will let researchers estimate not only main effects but also synergies that may occur in such research.

Some research indicates that psychosocial factors influence food choices in restaurants. Studies show that dining out is often motivated by convenience, pleasure, and value rather than health [132]. Also, social dynamics, including peer norms and dining companions, affect both the type and quantity of food ordered [133]. Stress and time scarcity also drive reliance on fast food [134], while digital platforms add new influences through promotions, loyalty programs, and targeted advertising [135].

Future research could go beyond documenting food choice behavior patterns to systematically test how psychosocial levers can be applied in restaurant settings. Priorities could include: (1) experiments that evaluate how framing choices, peoples’ identity, and customer norms can be harnessed to increase healthier meal selection without reducing satisfaction; (2) longitudinal studies that track how stress, habits, and social context interact with dependence on restaurants as a regular food source over time; (3) real-world trials on digital platforms to test nudges, reward structures, and personalized prompts that encourage healthier ordering; and (4) mixed-methods research that combines sales data with interviews or surveys to capture both behaviors and underlying motivations. This type of work would help move interventions from focusing only on menu content and labeling toward approaches that resonate with consumers’ lived experiences, thus perhaps increasing their effectiveness.

Some government policies have shown measurable effects on the association between restaurants and noncommunicable chronic disease rates. Looking ahead, new government policies could be designed and studied to address the evolving ways that restaurants contribute to noncommunicable chronic disease rates. For example, beyond kilocalorie labeling, governments could require “nutrient density labeling” that highlights the balance of kilocalories to fiber, protein, or vegetables on menu items, thus making it easier for consumers to identify meals that deliver more nutrition per kilocalorie. Policy pilot programs might also include tax credits or subsidies for restaurants that meet benchmarks for healthier default meals, such as those that offer fruit, water, or reduced-sodium items in children’s combination meals. This would mirror agricultural subsidies but apply them directly to menu design. At the community level, zoning policies could shift from simply limiting fast-food outlets to incentivizing the placement of healthier quick-service models in underserved areas, thus changing food environments from “food swamps” into “healthy-choice corridors.” With the rapid rise in delivery platforms that include traditional restaurants and ghost kitchen offerings, governments could also require digital menu standards. For example, require lower-calorie or lower-sodium options at the top of app interfaces or mandate clear sodium and added-sugar warnings online. Research on these novel approaches could test not only impacts on nutrient intake and chronic disease risk but also on restaurant economics, consumer satisfaction, and equity. This would help policymakers design interventions that balance health promotion with the economic sustainability of the restaurant industry.

The management of food allergies in restaurants continues to present major challenges. These are reported to be training gaps, communication failures, and cross-contact risks, all of which can contribute to severe reactions when consumers dine out. Research should test standardized allergy training programs and menu/POS disclosure systems that are integrated with recipe data. It could also evaluate validated cross-contact prevention methods, such as surface swabbing or designated preparation areas. Outcomes could include the accuracy of allergen information provided to guests, the incidence of documented reactions, and feasibility metrics such as training time and staff turnover [136].

Future nutrition research could also address the growing share of meals consumers obtain from nontraditional food-away-from-home sources, including food trucks and delivery-only restaurants (ghost kitchens). These formats are increasingly contributing to daily dietary intake, yet their nutritional characteristics and implications for public health remain underexplored [137,138]. Studies of mobile food vendors indicate that only a small fraction of menu items meet “healthy” criteria, and many operators cite low consumer demand or operational constraints as barriers to offering healthier options [137]. To fill these gaps, future research could examine nutrient composition, portion sizes, and consumer choices associated with these emerging dining formats. Research could also explore allergen-related issues, such as the presence and disclosure of common allergens and the potential for unintentional exposure. In addition, future review studies will be important to synthesize emerging findings and monitor how these evolving food service models influence diet quality, allergen exposure, and overall related public health outcomes over time.

### 5.4. Research Gaps and Suggestions for Future Food Science and Restaurant Research

Food safety remains both a continuing public health risk and a significant liability risk for restaurants, as outbreaks linked to ill workers, poor hygiene, or time–temperature control continue to occur. Yet while surveillance and inspection data consistently highlight these recurring food safety issues, related research on restaurant-partnered intervention trials that determine which solutions truly reduce risks in restaurant operations is lacking. Many restaurants still lack comprehensive illness-exclusion policies or feasible supports like paid sick leave, and although establishments with robust food safety management systems (FSMS) perform better, widespread adoption and evaluation of FSMS components remain limited [20,139].

A research priority is to evaluate multi-component employee-health programs that address the root causes of worker-related contamination. These could combine Food Code–consistent written illness policies, practical exclusion supports such as paid sick leave or shift-coverage plans, active managerial controls like routine symptom checks, and environmental supports such as touchless fixtures and handwashing prompts. Evidence from FDA’s norovirus risk assessment suggests that prompt worker exclusion, paired with strict hand hygiene and no bare-hand contact, significantly reduces transmission risk, providing a strong rationale for bundled trials [140]. Future studies could compare single interventions to bundled approaches, measure compliance behaviors and training adherence, and link them to both safety outcomes (e.g., complaints, incidents) and business outcomes (e.g., labor disruption, service times).

Another priority is food safety research on the away-from-home food preparation ecosystem, such as food trucks, online delivery platforms, curbside pickup, and ghost kitchens. These have all grown rapidly, but are understudied [141,142]. They need research trials that test such topics as cold-/hot-chain integrity through time–temperature logging, and tamper-evident and insulated packaging. While the U.S. Food and Drug Administration and the Conference for Food Protection have issued best-practice recommendations, related empirical evidence is still sparse. Partnered studies could assess the percentage of orders kept within safe temperature limits, track contamination or cross-contact, and evaluate impacts on throughput and customer experience [37].

Future research is needed to develop evidence-based frameworks for menu and restaurant design that both enhance dining pleasure and nudge consumers toward healthier and more sustainable choices. Each of the intersecting fields of molecular gastronomy, neurogastronomy, and gastrophysics brings a unique but complementary perspective to this goal. Molecular gastronomy focuses on the physical and chemical transformations of ingredients, providing chefs and food scientists with methods to improve texture, flavor intensity, and nutritional retention through precise control of cooking processes [143]. Neurogastronomy examines how the brain constructs flavor by integrating sensory signals, memories, and expectations, offering insight into how menu descriptions, presentation, and context can influence decision-making and satisfaction [49]. Researchers focused on gastrophysics examine how environmental and contextual factors such as lighting, sound, color, and plating can shape perception and behavior in real-world settings [53]. For example, a collaborative research project might explore how the chemical modification of plant-based proteins (molecular gastronomy) interacts with neural reward responses to umami (neurogastronomy) and how restaurant lighting and ambient sound can be adjusted to enhance both flavor perception and sustainable menu appeal (gastrophysics).

Current dietary trends and social movements such as veganism, farm-to-table dining, functional foods, and medically tailored menus present both opportunities and challenges for the future of food science and restaurant innovation. These movements reflect growing consumer awareness of health, sustainability, and ethics, yet they also demand scientific and technological advances to ensure that the foods offered meet nutritional, sensory, and environmental expectations. For example, from a food science perspective, developing plant-based alternatives that replicate the texture, flavor, and nutrient density of animal products remains a significant research challenge, requiring progress in protein chemistry, emulsification, and flavor compound development [143,144]. Similarly, farm-to-table operations must address issues of food safety, shelf stability, and nutrient retention in minimally processed ingredients through innovations in preservation and traceability technologies [145,146]. The rise of functional foods, that is, those designed to promote specific health outcomes necessitates that research into ingredient bioavailability and sensory optimization is peformed to balance efficacy with consumer acceptance [147]. Also, offering medically tailored menus for people with such conditions as type II diabetes in restaurant settings creates a complex set of challenges for chefs and food scientists alike [148]. They must reconcile therapeutic nutritional goals with appealing taste and presentation. Future interdisciplinary research is needed to align culinary creativity with public health and sustainability objectives.

Food fraud is the deliberate substitution, addition, tampering, or misrepresentation of food products for economic gain [149]. Restaurant operators often depend on complex, globalized supply chains to obtain ingredients, which increases the likelihood of unknowingly purchasing fraudulent products. Serving these items can expose operators to reputational damage, financial losses, and potential legal liability [150].

Food science research can play a critical role in helping restaurants address food fraud. Analytical advances such as DNA barcoding, isotope ratio mass spectrometry, and spectroscopic fingerprinting have shown promise in detecting species substitution in fish and meat products, authenticating olive oil and wine, and identifying adulteration in herbs and spices [151,152]. However, these methods are typically used at the manufacturing or regulatory level rather than being adapted to restaurant procurement systems. Future research could focus on developing rapid, low-cost, and portable detection tools that can be integrated into restaurant supply chains or used by distributors and buyers to verify authenticity at point-of-purchase.

Environmental sustainability is another pressing concern for the restaurant industry. Life cycle assessments indicate that ingredient substitution, such as shifting from animal proteins to legumes or hybrid formulations, can substantially reduce a meal’s carbon footprint [153]. However, restaurant research has rarely examined how such substitutions perform in terms of flavor, customer acceptance, or operational efficiency. Future work could evaluate strategies for lowering greenhouse gas emissions and water use while maintaining customer satisfaction. In addition, packaging innovations also represent fertile ground for food science research. The rise in take-out and delivery has accelerated demand for packaging that is both sustainable and able to preserve food quality. Biodegradable, recyclable, or edible materials may provide alternatives, but more research on these topics is needed to test performance and cost-effectiveness in restaurant environments [154].

Food science research needs to address industry resilience and crisis preparedness, especially in light of recent supply-chain disruptions that were caused by global events that exposed the fragility of just-in-time food systems [155,156,157]. Restaurants often operate with thin margins and rely heavily on predictable ingredient flows, making them vulnerable to shortages and price shocks [158]. Research could evaluate how preservation technologies such as high-pressure processing and pulsed electric fields extend the shelf life of perishables, thus reducing the need for frequent deliveries [159,160]. Another promising direction could be the development of modular, flexible recipes that enable ingredient substitution without compromising flavor, nutrition, or guest satisfaction. For example, designing sauces, bases, or side dishes that can be made using alternative ingredients, depending on the availability of various seasonal or locally available produce [161].

### 5.5. Study Limitations

There are several limitations in this study that need to be acknowledged. While Scopus and WoS databases cover a large portion of published research articles, other sources like ProQuest and Google Scholar might contain additional relevant papers. Furthermore, the study focused on journal articles, excluding books, dissertations, and conference proceedings, which could be explored in future research. Potential methodological limitations common to bibliometric analyses are acknowledged. Specifically, the use of VOSviewer clustering thresholds, choice of co-occurrence measures, and normalization techniques can influence the structure and interpretation of networks, and results are inherently time-bound, reflecting only the state of the literature up to the retrieval date.

## 6. Conclusions

This study is the first bibliometric analysis to explore research that has focused on nutrition and food science topics related to the restaurant industry. It uncovered a significant imbalance in research focus. Nutrition-related studies dominate the field, with most focusing on fast food consumption, menu labeling, consumer behavior, and associations with noncommunicable chronic disease and its risk factors. By contrast, restaurant-focused food science research remains comparatively limited, concentrating largely on food safety topics and reuse opportunities for used frying oil. Additionally, the overwhelming focus on fast food has left significant research gaps in emerging sectors such as fast casual and fine dining, despite their increasing influence on consumer dietary patterns.

Future studies should move beyond short-term, cross-sectional designs and incorporate longitudinal approaches to better capture how policy interventions, such as kilocalorie labeling and reformulation incentives, affect restaurant offerings and consumer food choices over time. Interdisciplinary research is recommended to explore the acceptability of more sustainable ingredient substitutions and the effects of restaurant practices on allergen management. Additionally, restaurant-partnered interventions that balance public health goals with operational and environmental considerations are suggested. By broadening the scope of inquiry and applying more rigorous study designs, future scholarship can provide actionable evidence to help restaurants contribute to healthier diets and more sustainable practices.

## Figures and Tables

**Figure 1 nutrients-17-03490-f001:**
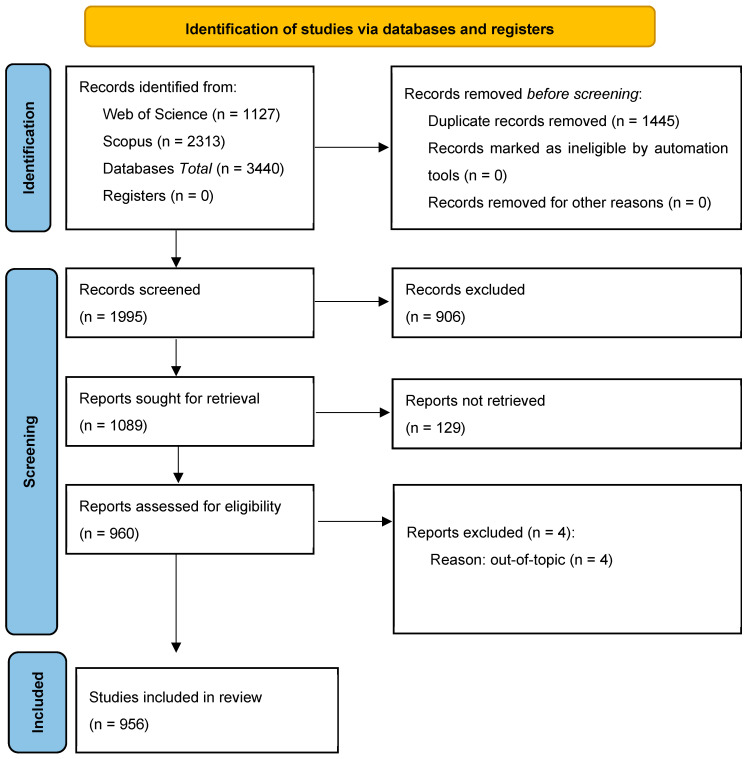
PRISMA flow diagram illustrating the article selection process for this bibliometric review of nutrition and food science-related research in the restaurant industry. The PRISMA flow diagram is licensed under CC BY 4.0. To view a copy of this license, visit https://creativecommons.org/licenses/by/4.0/ (accessed on 30 November 2024).

**Figure 2 nutrients-17-03490-f002:**
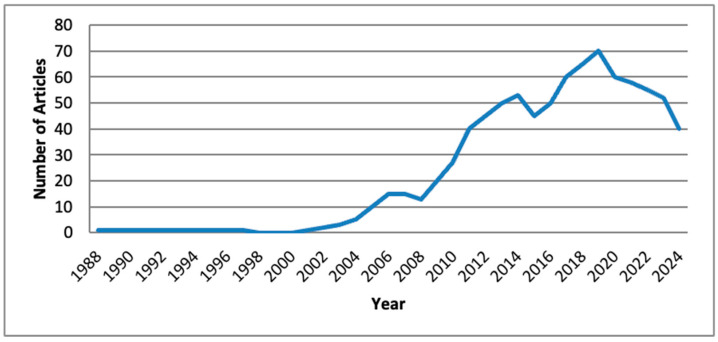
Annual publication trends in restaurant-related nutrition research (*n* = 824).

**Figure 3 nutrients-17-03490-f003:**
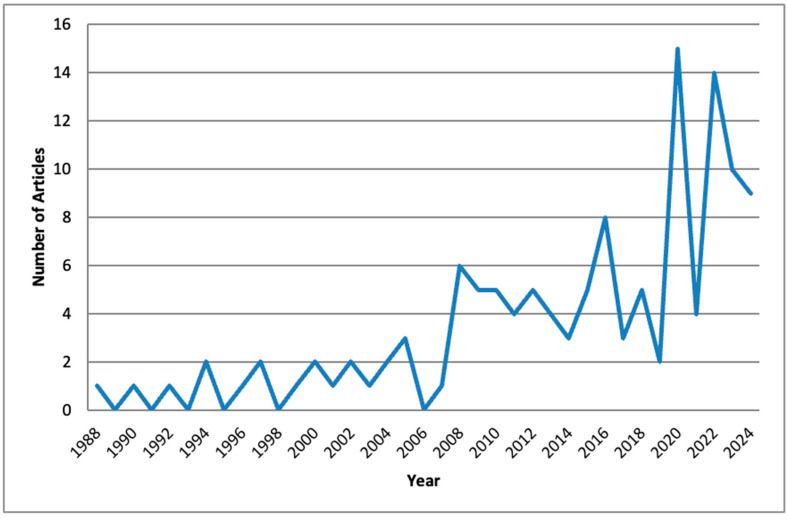
Annual publication trends in restaurant-related food science research (*n* = 132).

**Figure 4 nutrients-17-03490-f004:**
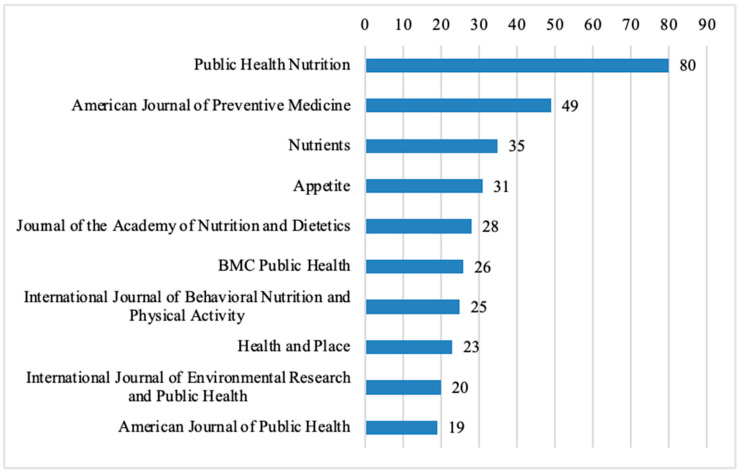
Top 10 journals publishing the most articles on nutrition research related to restaurants (*n* = 824).

**Figure 5 nutrients-17-03490-f005:**
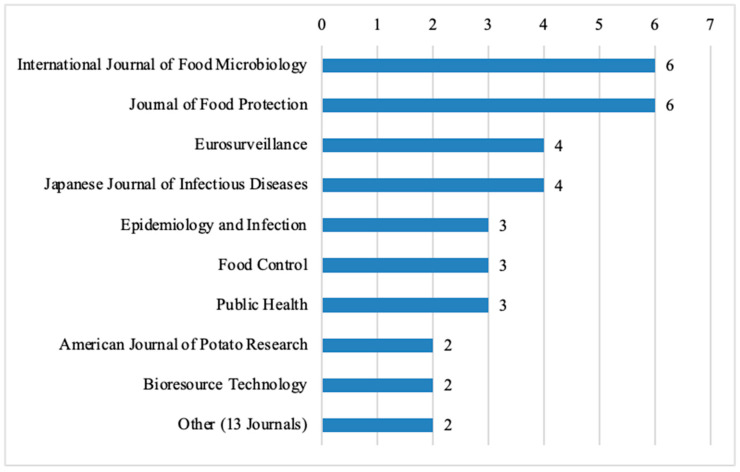
Top 10 journals publishing the most articles on food science research related to restaurants (*n* = 132).

**Figure 6 nutrients-17-03490-f006:**
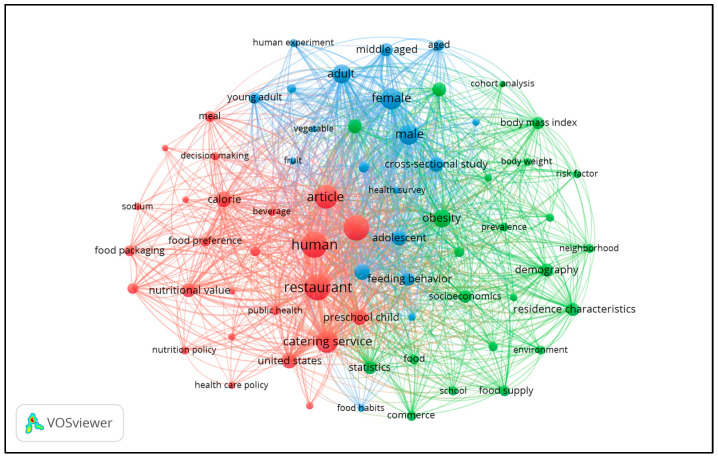
A lexical network visualization map displaying the co-occurrence of all keywords within the documents (*n* = 824) focused on nutrition that met the defined threshold for this study (*n* = 50). Each color indicates a distinct group of interconnected nodes.

**Figure 7 nutrients-17-03490-f007:**
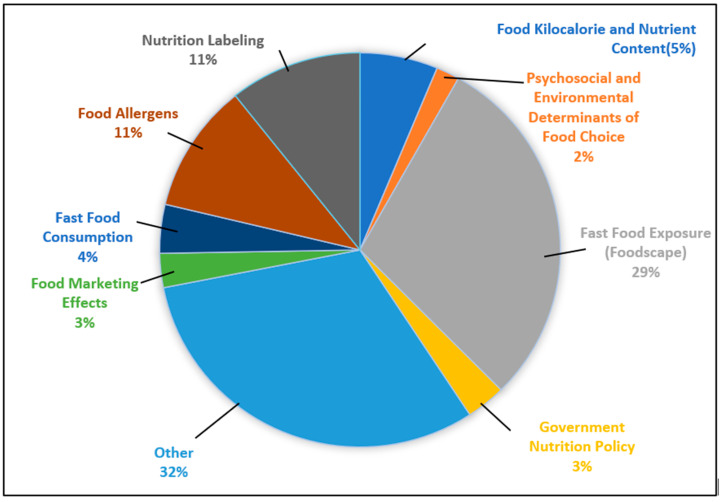
Explanatory variables in studies about nutrition and restaurants.

**Figure 8 nutrients-17-03490-f008:**
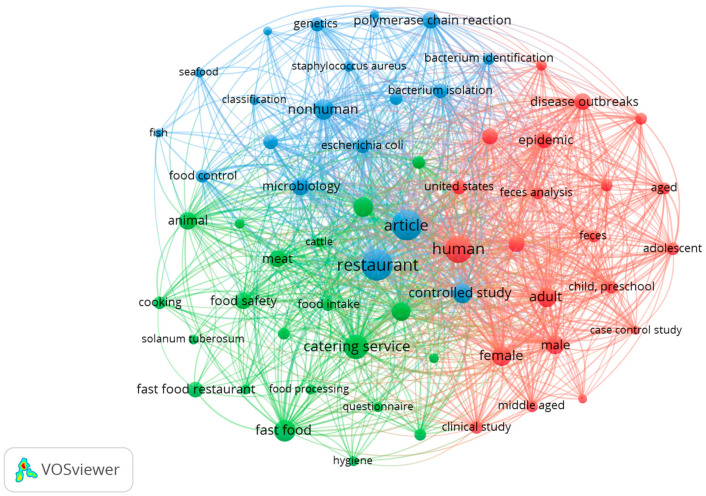
A network visualization map displaying the co-occurrence of all keywords within the documents (*n* = 132) focused on food science that met the defined threshold for this study (*n* = 8). Each color indicates a distinct group of interconnected nodes.

**Figure 9 nutrients-17-03490-f009:**
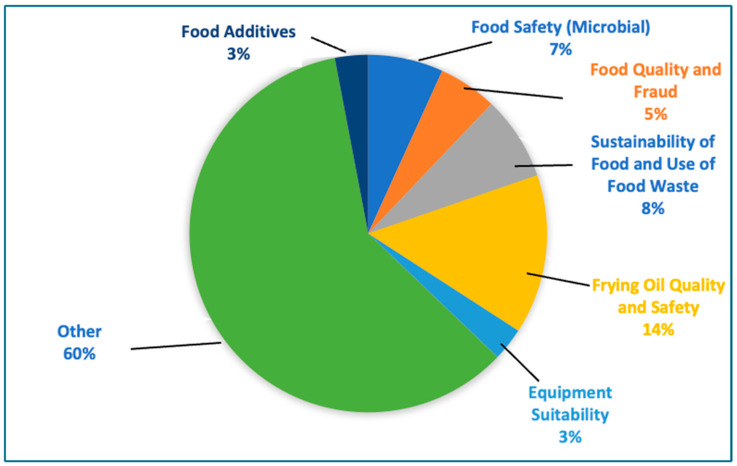
Explanatory variables in studies about restaurants and food science.

**Table 1 nutrients-17-03490-t001:** Top ten most cited articles in nutrition and restaurant research.

No.	Citations	Citations per Year	Title	Publication	Restaurant Category
1	1541	73.38	An economic analysis of adult obesity: Results from the behavioral risk factor surveillance system	Journal of Health Economics	Fast food and full service
2	1204	50.17	Fast food restaurant use among adolescents: Associations with nutrient intake, food choices and behavioral and psychosocial variables	International Journal of Obesity	Fast food
3	1019	48.52	Fast food consumption of U.S. adults: Impact on energy and nutrient intakes and overweight status	Journal of the American College of Nutrition	Fast food
4	849	33.96	Fast food restaurant use among women in the pound of prevention study: Dietary, behavioral and demographic correlates	International Journal of Obesity	Fast food
5	736	46.00	Proximity of fast-food restaurants to schools and adolescent obesity	American Journal of Public Health	Fast food
6	794	41.79	Are fast food restaurants an environmental risk factor for obesity?	International Journal of Behavioral Nutrition and Physical Activity	Fast food
7	687	34.35	Clustering of fast-food restaurants around schools: A novel application of spatial statistics to the study of food environments	American Journal of Public Health	Fast food
8	683	42.69	Calorie labeling and food choices: A first look at the effects on low-income people in New York City	Health Affairs	Fast food
9	582	27.71	Neighborhood playgrounds, fast food restaurants, and crime: Relationships to overweight in low-income preschool children	Preventive Medicine	Fast food
10	556	39.71	Fast food restaurants and food stores longitudinal associations with diet in young to middle-aged adults: The CARDIA study	Archives Internal Medicine	Fast food

**Table 2 nutrients-17-03490-t002:** Top ten most cited articles in food science and restaurant research.

No.	Citations	Citations per Year	Title	Publication	Restaurant Category
1	642	23.8	Molecular epidemiology of ‘norwalk-like viruses’ in outbreaks of gastroenteritis in the United States	Journal of Infectious Diseases	Restaurant unspecified
2	221	17.0	Presence of faecal coliforms, Escherichia coli and diarrheagenic E. coli pathotypes in ready-to-eat salads, from an area where crops are irrigated with untreated sewage water	International Journal of Food Microbiology	Restaurant unspecified
3	216	13.5	The Real maccoyii: Identifying Tuna Sushi with DNA Barcodes--Contrasting Characteristic Attributes and Genetic Distances	PloS One	Restaurant unspecified
4	216	10.8	A review of the needs and current applications of hazard analysis and critical control point (HACCP) system in foodservice areas	Food Control	Restaurant unspecified
5	174	29.0	A life cycle assessment of the environmental impacts of a beef system in the USA	The International Journal of Life Cycle Assessment	Restaurant unspecified
6	120	6.7	Reconsidering the effects of monosodium glutamate: A literature review	Journal of the American Academy of Nurse Practitioners	Restaurant unspecified
7	114	9.5	Cradle to retailer or quick service restaurant gate life cycle assessment of chicken products in Australia	Journal of Cleaner Production	Fast-food
8	114	19.0	Anthropogenic gadolinium in tap water and in tap water-based beverages from fast-food franchises in six major cities in Germany	Science of the Total Environment	Fast-food
9	109	18.2	Evaluating the rancidity and quality of discarded oils in fast food restaurants	Food Science & Nutrition	Fast-food
10	103	9.4	Effect of green tea extract and microwave pre-cooking on the formation of heterocyclic aromatic amines in fried chicken meat products	Food Research International	Restaurant unspecified

**Table 3 nutrients-17-03490-t003:** The interplay between restaurant category and food domain.

The Food Domains in Restaurant Research	Restaurant Category				
	Fast Food	Fast Casual	Casual Dining	Fine Dining	Restaurant Unspecified
Nutrition ○ Menu labelling ○ Offering healthy alternatives ○ Food marketing effects	√		√		√
Food Science ○ Food safety ○ Food waste use ○ Food sustainability	√		√	√	√

Note: Check marks indicate that at least one article focused on the particular restaurant category.

## Data Availability

Not applicable.

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
