# Peer review of "Intersection of Nutrition, Food Science, and Restaurant Research"

_nutrients, 2025, doi:10.3390/nu17213490_

Round 1
Reviewer 1 Report
Comments and Suggestions for Authors
The paper highlights differences from related research, as it focuses on different dimensions of restaurant services, specifically from a dietary perspective. The article could introduce ideas that might stimulate specialists in the field to improve menu design, or design and develop applications that instantly reveal the nutritional data of dishes, or provide 3D food printing. The paper could make a valuable contribution to the area of supporting entrepreneurs in the field, enabling them to improve specific client services in line with current perspectives.
The article is well-documented and comprehensive. The authors could consider adding current trends data regarding gastrophysics. Along with the molecular gastronomy that they discussed, it could complete the picture of possible examples of collaborations between researchers and chefs.
Information regarding the challenges posed by current trends and innovations in the field, such as those related to social movements like veganism, farm-to-table, functional food, and menus tailored to medical needs, could also be included if available in the literature.
Author Response
Dear Reviewer,
We sincerely appreciate the time and thoughtful feedback you provided to help us enhance the quality of our bibliometric review on nutrition and food science topics related to the restaurant industry. Our goal in conducting this study was to offer readers an overview of research trends in this area and to highlight key gaps that warrant further investigation. Your comments have helped us with our goal.
Comment 1: The article is well-documented and comprehensive. The authors could consider adding current trends data regarding gastrophysics. Along with the molecular gastronomy they discussed, this could complete the picture of possible collaborations between researchers and chefs.
Response 1: This is an excellent point and an oversight on our part. We have added sections in the Literature Review and Discussion to address this shortcoming. Please see below.
Literature Review
The research area called neurogastronomy focuses on how the brain creates and interprets flavor, emphasizing that such perception involves complex multisensory and cognitive processes rather than simple responses to food stimuli [49]. As discussed by Spence [50], research in this area explores how sensory cues, such as aroma, texture, visual presentation, and contextual information, such as pricing, may affect consumers’ neural responses during eating. For example, Chen, Papies, and Barsalou [51] identified a “core eating network” in the brain that integrates sensory and cognitive information to guide feeding behavior. Studies have also shown that brand and price cues can modulate neural activation in reward-related regions, leading participants to rate identical wines as tasting better when labeled with higher prices [52].
A term combining gastronomy and psychophysics is gastrophysics which is an area of research that investigates how environmental and contextual factors influence the perception and enjoyment of food [53]. It emphasizes that dining is a multisensory ex-perience shaped not only by the chemical composition of food but also by cues found in the environment where the food will be consumed [50]. Research in this area has shown, for example, that the color and shape of tableware can alter perceived sweetness or saltiness [54], and that background music can modulate taste perception and emotional responses to food [55]. Similarly, artistic plating has been found to increase both enjoyment and willingness to pay, even when the ingredients remain the same [56].
Discussion
Future research is needed to develop evidence-based frameworks for menu and restaurant design that both enhance dining pleasure and nudge consumers toward healthier and more sustainable choices. Each of the intersecting fields of molecular gastronomy, neurogastronomy, and gastrophysics brings a unique but complementary perspective to this goal. Molecular gastronomy focuses on the physical and chemical transformations of ingredients, providing chefs and food scientists with methods to improve texture, flavor intensity, and nutritional retention through precise control of cooking processes [143]. Neurogastronomy examines how the brain constructs flavor by integrating sensory signals, memories, and expectations, offering insight into how menu descriptions, presentation, and context can influence decision-making and satisfaction [49]. Researchers focused on gastrophysics examine how environmental and contextual factors such as lighting, sound, color, and plating can shape perception and behavior in real-world settings [53]. For example, a collaborative research project might explore how the chemical modification of plant-based proteins (molecular gastronomy) interacts with neural reward responses to umami (neurogastronomy) and how restaurant lighting and ambient sound can be adjusted to enhance both flavor perception and sustainable menu appeal (gastrophysics).
Comment 2: Information regarding the challenges posed by current trends and innovations in the field, such as those related to social movements like veganism, farm-to-table, functional food, and menus tailored to medical needs, could also be included if available in the literature.
Response 2: Yes we agree this will be an important addition to our discussion section. Below is what we have added.
Discussion
Current dietary trends and social movements such as veganism, farm-to-table dining, functional foods, and medically tailored menus present both opportunities and challenges for the future of food science and restaurant innovation. These movements reflect growing consumer awareness of health, sustainability, and ethics, yet they also demand scientific and technological advances to ensure that the foods offered meet nutritional, sensory, and environmental expectations. For example, from a food science perspective, developing plant-based alternatives that replicate the texture, flavor, and nutrient density of animal products remains a significant research challenge, requiring progress in protein chemistry, emulsification, and flavor compound development [143,144]. Similarly, farm-to-table operations must address issues of food safety, shelf stability, and nutrient retention in minimally processed ingredients through innovations in preservation and traceability technologies [145,146]. The rise of functional foods, that is, those designed to promote specific health outcomes necessitates that research into ingredient bioavailability and sensory optimization is peformed to balance efficacy with consumer acceptance [147]. Also, offering medically tailored menus for people with such conditions as type II diabetes in restaurant settings creates a complex set of challenges for chefs and food scientists alike [148]. They must reconcile therapeutic nutritional goals with appealing taste and presentation. Future interdisciplinary research is needed to align culinary creativity with public health and sustainability objectives.
Thank you!
Reviewer 2 Report
Comments and Suggestions for Authors
This study provides an interesting exploration of the intersection between restaurants, nutrition, and public health outcomes.
The choice to limit the restaurant categories to fast food, fast casual, casual dining, and fine dining seems logical, but it's not clear why other categories—like casual eateries, food trucks, or the growing trend of delivery-only restaurants (ghost kitchens)—weren’t included. These formats are becoming more prominent in shaping food access, consumer behavior, and dietary patterns.
The study rightly excludes business operations, but this creates a gap in understanding how restaurant economics (pricing, portion sizes, or marketing) influence public health outcomes. Often, restaurant success and food offerings are closely tied to pricing strategies and operational models. Excluding these factors could limit the understanding of how broader economic dynamics affect dietary behaviors.
he paper notes that nutrition-related studies dominate, particularly linking fast food to obesity. While this focus is important, there’s little mention of how broader socio-cultural or environmental factors influence food choices in restaurants. Nutrition alone, while essential, may oversimplify the problem.
The fact that food science research is more limited, particularly focusing on food safety and degraded frying oil, indicates a potential underrepresentation of critical areas such as the environmental impact of restaurant operations or sustainability in food sourcing.
Furthermore, the focus on menu labeling policies is appropriate, but a more detailed discussion of their actual effectiveness would be beneficial. The study doesn’t address how different regions or populations may respond to these interventions. Menu labeling may be a good example of a "one-size-fits-all" policy that doesn’t account for diverse cultural preferences or differing levels of health literacy.
The critique of short-term, cross-sectional studies is valid. However, this could be expanded upon by identifying why longitudinal studies are so important in this field. It's easy to imagine that a menu labeling intervention might lead to short-term changes in consumer behavior, but does this persist over time? Do people become complacent, or do they simply change behaviors in response to external cues (like calorie counts on menus)?
The paper could discuss how people's food choices are often more influenced by psychological or emotional factors than rational decision-making, which could be especially important for the restaurant industry.
The focus on allergen risks and training programs is a very relevant public health concern, but it could be critiqued for being somewhat narrow. Allergen risks are only one part of food safety; the broader issue of foodborne illnesses or contamination could also be explored in restaurant contexts, which would further support public health outcomes
Author Response
Dear Reviewer,
We sincerely appreciate the time and thoughtful feedback provided to help us enhance the quality of our bibliometric review on nutrition and food science topics related to the restaurant industry. Our goal in conducting this study was to offer readers an overview of research trends in this area and to highlight key gaps that warrant further investigation.
Comment 1: The choice to limit the restaurant categories to fast food, fast casual, casual dining, and fine dining seems logical, but it's not clear why other categories—like casual eateries, food trucks, or the growing trend of delivery-only restaurants (ghost kitchens)—weren’t included. These formats are becoming more prominent in shaping food access, consumer behavior, and dietary patterns.
Response 1: We appreciate the reviewer’s insightful comment regarding the exclusion of other restaurant categories such as casual eateries, food trucks, and delivery-only restaurants (ghost kitchens). We agree that these are important and rapidly evolving sectors that are reshaping food access, consumer behavior, and dietary patterns.
Our review focused on restaurant types (i.e., fast food, fast casual, casual dining, and fine dining) that have an established body of research specifically related to food science and nutrition. While we could have broadened our scope to include nutrition and food science topics focused on all types of food prepared away from home, such as food trucks, convenience stores, cafeterias, institutional dining, street food vendors, and delivery-only kitchens. But our intention was to maintain a clear focus on restaurant settings with sufficient existing literature to allow for meaningful bibliometric analysis about nutrition and food science topics.
To further explore the reviewer’s valuable suggestion, we conducted a search in the Scopus database using the term “ghost kitchen(s)” in the title, keywords, or abstract. This search yielded 19 papers as of Oct. 17th, 2025. A quick review revealed that none of these papers focused specifically on nutrition, and one addressed a topic related to food science. Notably, six of these papers were published in 2025, after our data extraction period. Thus, there is a trend in increased publications related to ghost kitchens, but not ghost kitchens and nutrition and food science. We searched in Scopus for Food Trucks and Mobile Food Vendors and found 9 papers. Several of these are focused on food safety and nutrition topics.
We recognize the importance of these emerging food prepared away from home formats and have enhanced our recommendations for future research to encourage the inclusion of food trucks, casual eateries, and ghost kitchens in studies examining nutrition and food science in the restaurant industry.
Discussion additions:
Line 782
With the rapid rise of delivery platforms that include traditional restaurants and ghost kitchen offerings, governments could also require digital menu standards.
Line 798
Future nutrition research could also address the growing share of meals consumers obtain from nontraditional food-away-from-home sources, including food trucks and delivery-only restaurants (ghost kitchens). These formats are increasingly contributing to daily dietary intake, yet their nutritional characteristics and implications for public health remain underexplored [137,138]. Studies of mobile food vendors indicate that only a small fraction of menu items meet “healthy” criteria, and many operators cite low consumer demand or operational constraints as barriers to offering healthier options [137]. To fill these gaps, future research could examine nutrient composition, portion sizes, and consumer choices associated with these emerging dining formats. Research could also explore allergen-related issues, such as the presence and disclosure of common allergens and the potential for unintentional exposure. In addition, future review studies will be important to synthesize emerging findings and monitor how these evolving food service models influence diet quality, allergen exposure, and overall related public health outcomes over time.
Line 828 (we modified this section to include food trucks)
Another priority is food safety research on the away-from-home food preparation ecosystem, such as food trucks, online delivery platforms, curbside pickup, and ghost kitchens. All of these have grown rapidly, but are understudied (Global Market Insights, 2024, Verified Market Reports, 2024). These all need research trials that study such topics as cold-/hot-chain integrity through time–temperature logging, and tamper-evident and insulated packaging. While the U.S. Food and Drug Administration and the Conference for Food Protection have issued best-practice recommendations, related empirical evidence is still sparse. Partnered studies could assess the percentage of orders kept within safe temperature limits, track contamination or cross-contact, and evaluate impacts on throughput and customer experience [37].
Comment 2: The study rightly excludes business operations, but this creates a gap in understanding how restaurant economics (pricing, portion sizes, or marketing) influence public health outcomes. Often, restaurant success and food offerings are closely tied to pricing strategies and operational models. Excluding these factors could limit the understanding of how broader economic dynamics affect dietary behaviors.
Response 2: Thank you for pointing this out. Our methods section didn’t clearly indicate that these types of business topics were indeed included in this study. We have clarified this in the methods section. Also, Figure 7 indicates that marketing studies related to nutrition made up 3% of the nutrition-related papers.
Line 286 (Methods addition)
Only articles addressing nutrition or food science-related aspects of the restaurant industry were included. Articles related to the business themes of the restaurant (i.e., organizational behavior, consumer behavior, and finance) sector or lacking full-text were not selected [2]. However, articles with business topics related to food science or nutrition, such as pricing and marketing effects on food choice, were included. Ultimately, 956 journal articles were used for this bibliometric analysis.
Comment 3: The paper notes that nutrition-related studies dominate, particularly linking fast food to obesity. While this focus is important, there’s little mention of how broader socio-cultural or environmental factors influence food choices in restaurants. Nutrition alone, while essential, may oversimplify the problem.
Response 3: Yes, indeed, thank you for the reminder that the causes of the global obesity epidemic are multifaceted. If we understand this comment correctly, we have addressed in our Literature Review that affordability, rising incomes, urbanization and marketing are all associated with the rise in obesity. Also, Figure 7 indicates psychosocial and environmental determinants of food choice have been studied in 2% of the articles focused on nutrition research.
Line 118 (Literature Review)
At the same time, fast food consumption has surged globally, driven by rapid urbanization, economic growth, and aggressive marketing strategies. A recent systematic review found that increased availability and affordability of fast food have contributed to shifts in eating patterns and have been linked to rising rates of diet-related chronic diseases in both high-income and low- to middle-income countries [24]. Diets are changing with rising incomes and urbanization, specifically, people are consuming more animal-based foods, sugar, fats and oils, refined grains, and processed foods. This global shift, often referred to as the nutrition transition, represents a critical challenge for public health and food policy.
Line 497 (Results)
Finally, psychosocial and environmental determinants of food choice (2%) addressed individual and contextual factors, such as social norms, stress, convenience, and time constraints that affect how people make food decisions in restaurant environments.
Line 751 (Discussion)
Some research indicates that psychosocial factors influence food choices in restaurants. Studies show that dining out is often motivated by convenience, pleasure, and value rather than health [126]. Also, social dynamics, including peer norms and dining companions, affect both the type and quantity of food ordered [127]. Stress and time scarcity also drive reliance on fast food [128], while digital platforms add new influences through promotions, loyalty programs, and targeted advertising [129].
Future research could go beyond documenting food choice behavior patterns to systematically test how psychosocial levers can be applied in restaurant settings. Priorities could include: (1) experiments that evaluate how framing choices, peoples’ identity, and customer norms can be harnessed to increase healthier meal selection without reducing satisfaction; (2) longitudinal studies that track how stress, habits, and social context interact with dependence on restaurants as a regular food source over time; (3) real-world trials on digital platforms to test nudges, reward structures, and personalized prompts that encourage healthier ordering; and (4) mixed-methods research that combines sales data with interviews or surveys to capture both behaviors and underlying motivations. This type of work would help move interventions from focusing only on menu content and labeling toward approaches that resonate with consumers’ lived experiences, thus perhaps increasing their effectiveness.
Comment 4: The fact that food science research is more limited, particularly focusing on food safety and degraded frying oil, indicates a potential underrepresentation of critical areas such as the environmental impact of restaurant operations or sustainability in food sourcing.
Response 4: Thank you for pointing this out. We agree that the environmental impact of restaurant operations and the sustainability of food sourcing are important topics related to food science. We found that about 8% of the articles focused on food science topics were within these topics. Topics related to the sustainability of restaurant operations, such as energy and water usage, were considered business topics (not nutrition or food science related) and not included in this review. However, they were covered in our previous review of business-related topics and restaurants. The areas in this paper about nutrition and food science as related to sustainability are below.
Line 551 (Results Table 9)
The sustainability of food and use of food waste theme accounted for 8% of the studies and addressed strategies such as converting food waste into animal feed or biodiesel and applying life-cycle assessment tools to restaurant operations [68,72,73].
Line 892 (Discussion)
Environmental sustainability is another pressing concern for the restaurant industry. Life cycle assessments indicate that ingredient substitution, such as shifting from animal proteins to legumes or hybrid formulations, can substantially reduce a meal’s carbon footprint [137]. However, restaurant research has rarely examined how such substitutions perform in terms of flavor, customer acceptance, or operational efficiency. Future work could evaluate strategies for lowering greenhouse gas emissions and water use while maintaining customer satisfaction. In addition, packaging innovations also represent fertile ground for food science research. The rise of take-out and delivery has accelerated demand for packaging that is both sustainable and able to preserve food quality. Biodegradable, recyclable, or edible materials may provide alternatives, but more research on these topics is needed to test performance and cost-effectiveness in restaurant environments [138].
Comment 5: Furthermore, the focus on menu labeling policies is appropriate, but a more detailed discussion of their actual effectiveness would be beneficial. The study doesn’t address how different regions or populations may respond to these interventions. Menu labeling may be a good example of a "one-size-fits-all" policy that doesn’t account for diverse cultural preferences or differing levels of health literacy.
Response 5: Thank you, we have added the following to address this important point.
Line 713 (Discussion addition)
Such research should account for the demographic characteristics of the customers being studied, as cultural backgrounds and regional contexts may lead to differing responses to these interventions.
Comment 6: The critique of short-term, cross-sectional studies is valid. However, this could be expanded upon by identifying why longitudinal studies are so important in this field. It's easy to imagine that a menu labeling intervention might lead to short-term changes in consumer behavior, but does this persist over time? Do people become complacent, or do they simply change behaviors in response to external cues (like calorie counts on menus)?
Response 6: Yes, thank you. This needs to be pointed out. We added the following to the discussion section.
Line 663 (Discussion addition)
Longitudinal studies are essential in this area because they can determine whether the effects of menu labeling and similar interventions are sustained over time, helping to distinguish between temporary behavioral shifts driven by novelty or external cues and lasting changes in dietary decision-making patterns.
Comment 7: The paper could discuss how people's food choices are often more influenced by psychological or emotional factors than rational decision-making, which could be especially important for the restaurant industry.
Response 7: Yes, we agree that food choice is influenced by many factors other than rational decision-making. Please see our response to Comment 3 above.
Comment 8:
The focus on allergen risks and training programs is a very relevant public health concern, but it could be critiqued for being somewhat narrow. Allergen risks are only one part of food safety; the broader issue of foodborne illnesses or contamination could also be explored in restaurant contexts, which would further support public health outcomes
Response 8: Thank you for this observation. Yes, we agree. We found that about 7% of the papers focused on a food science topic that examined the microbial food safety of restaurant offerings. See Figure 9. We also addressed the need for future food safety research and food fraud research in the Discussion section.
Line 554 (Results)
Food safety (microbial) represented 7% of the dataset and primarily focused on the contamination risks associated with ingredients in restaurants and carried by employees. Studies investigated the presence of foodborne pathogens such as Escherichia coli, Salmonella spp., and Clostridium perfringens, which are commonly introduced through poor hygiene, cross-contamination, or improper temperature control [79,80]. Research also evaluated preventive measures in foodservice environments, such as kitchen sanitation practices and cold chain management in minimizing microbial hazards [81–83].
Line 813 (Discussion)
Food safety remains both a continuing public health risk and a significant liability risk for restaurants, as outbreaks linked to ill workers, poor hygiene, or time–temperature control continue to occur. Yet while surveillance and inspection data consistently highlight these recurring food safety issues, related research on restaurant-partnered intervention trials that determine which solutions truly reduce risks in restaurant operations is lacking. Many restaurants still lack comprehensive illness-exclusion policies or feasible supports like paid sick leave, and although establishments with robust food safety management systems (FSMS) perform better, widespread adoption and evaluation of FSMS components remain limited [20,139].
A research priority is to evaluate multi-component employee-health programs that address the root causes of worker-related contamination. These could combine Food Code–consistent written illness policies, practical exclusion supports such as paid sick leave or shift-coverage plans, active managerial controls like routine symptom checks, and environmental supports such as touchless fixtures and handwashing prompts. Evidence from FDA’s norovirus risk assessment suggests that prompt worker exclusion, paired with strict hand hygiene and no bare-hand contact, significantly reduces transmission risk, providing a strong rationale for bundled trials [140]. Future studies could compare single interventions to bundled approaches, measure compliance behaviors and training adherence, and link them to both safety outcomes (e.g., complaints, incidents) and business outcomes (e.g., labor disruption, service times).
Another priority is food safety research on the away-from-home food preparation ecosystem, such as food trucks, online delivery platforms, curbside pickup, and ghost kitchens. These have all grown rapidly, but are understudied [141,142]. They need research trials that test such topics as cold-/hot-chain integrity through time–temperature logging, and tamper-evident and insulated packaging. While the U.S. Food and Drug Administration and the Conference for Food Protection have issued best-practice recommendations, related empirical evidence is still sparse. Partnered studies could assess the percentage of orders kept within safe temperature limits, track contamination or cross-contact, and evaluate impacts on throughput and customer experience [37].
Line 896 (Discussion)
Food fraud is the deliberate substitution, addition, tampering, or misrepresentation of food products for economic gain [133]. Restaurant operators often depend on complex, globalized supply chains to obtain ingredients, which increases the likelihood of unknowingly purchasing fraudulent products. Serving these items can expose operators to reputational damage, financial losses, and potential legal liability [134].
Food science research can play a critical role in helping restaurants address food fraud. Analytical advances such as DNA barcoding, isotope ratio mass spectrometry, and spectroscopic fingerprinting have shown promise in detecting species substitution in fish and meat products, authenticating olive oil and wine, and identifying adulteration in herbs and spices [135,136]. However, these methods are typically used at the manufacturing or regulatory level rather than being adapted to restaurant procurement systems. Future research could focus on developing rapid, low-cost, and portable detection tools that can be integrated into restaurant supply chains or used by distributors and buyers to verify authenticity at point-of-purchase.